# Bioremediation of Contaminated Water: The Potential of Aquatic Plants *Ceratophyllum demersum* and *Pistia stratiotes* Against Toxic Bloom

**DOI:** 10.3390/toxins17100490

**Published:** 2025-10-02

**Authors:** Fatma Zohra Tamer, Hadjer Zaidi, Hichem Nasri, Larisa Lvova, Nada Nouri, Fateh Sedrati, Amina Amrani, Nassima Beldjoudi, Xi Li

**Affiliations:** 1Laboratory of Biodiversity and Ecosystems Pollution, Faculty of Life and Nature Sciences, University of Chadli Bendjedid, El-Tarf 36000, Algeria; ha.zaidi@univ-eltarf.dz (H.Z.); nasri-hicham@univ-eltarf.dz (H.N.); n.nouri@univ-eltarf.dz (N.N.); f.sedrati@univ-soukahras.dz (F.S.); amrani-amina@univ-eltarf.dz (A.A.); bn-cherchassocie-lrbpe@univ-eltarf.dz (N.B.); 2Department of Chemical Science and Technologies, Universty of Rome «Tor Vergata», Via della Ricerca Scientifica, 00133 Rome, Italy; 3Institute of Urban Environment (IUE), Chinese Academy of Sciences, 1799 jimei Road, Xiamen 361021, China; xli@iue.ac.cn

**Keywords:** cyanobacteria, microcystins, bioaccumulation, phytoremediation, *Pistia stratiotes*, *Ceratophyllum demersum*

## Abstract

Toxic cyanobacteria, including *Microcystis*, produce harmful toxins that affect aquatic ecosystems and human health. Biotreatment using macrophytes shows promise in mitigating these blooms. This study investigates the bioaccumulation dynamics and biochemical responses of two aquatic macrophytes, *Pistia stratiotes* and *Ceratophyllum demersum*, in removing microcystin from contaminated water. *P. stratiotes* showed high initial bioaccumulation rates with rapid microcystin uptake, which is effective for short-term bioremediation. *C. demersum* has shown stable bioaccumulation. Biochemical analyses have revealed the activation of plant antioxidant defenses, with both macrophytes showing an increase in carotenoids, glutathione (GSH), and antioxidant enzymes such as superoxide dismutase (SOD) and glutathione-S-transferase (GST) concentrations. In particular, *C. demersum* has maintained higher antioxidant levels, contributing to its sustained capacity and resilience. Fluctuations in malondialdehyde (MDA) indicated oxidative stress, with *P. stratiotes* managing such stress through its defenses. Principal Component Analysis (PCA) supports these findings: Pistia’s first two components explained 25.09% and 20.71% of the variance, with Carotenoid and Chl contributing strongly to PC1, and MDA and GST influencing both components. For *C. demersum*, PC1 and PC2 explained 21.79% and 19.78% of the variance, with Carotenoid and Chl a being major contributors, while SOD and GSH played significant roles in sample differentiation. Integrating both plants into bioremediation strategies could optimize microcystin removal: *P. stratiotes* offers rapid initial detoxification, while *C. demersum* ensures continuous, long-term remediation. This combined approach enhances the efficiency and sustainability of phytoremediation. Future research should optimize environmental conditions and explore synergistic effects among multiple plant species for more effective and sustainable bioremediation solutions.

## 1. Introduction

Microcystins, potent hepatotoxins produced by cyanobacteria such as *Microcystis*, pose a significant threat to human health and water quality worldwide. Exposure to microcystins can lead to severe health issues, including liver damage, gastrointestinal disturbances, and even tumor promotion [1]. These toxins are prevalent in freshwater ecosystems affected by eutrophication, resulting from nutrient pollution and global changes. The widespread occurrence of harmful algal blooms (HABs) exacerbates the risk of microcystin contamination, making it a pressing public health concern [2]. Understanding the mechanisms by which microcystins affect biological systems is essential to fully grasp the severity of their impact. Microcystins exert toxic effects by inhibiting protein phosphatases 1 and 2A, leading to the disruption of cellular functions and promoting oxidative stress. This oxidative stress can cause lipid peroxidation, DNA damage, and apoptosis, further highlighting the urgency of finding effective solutions to mitigate these toxins [3,4].

Physicochemical methods for mitigating microcystin contamination, such as chemical treatments and mechanical filtration, are often costly and may have adverse ecological impacts. These methods can also lead to the release of secondary pollutants and require frequent maintenance, making them less feasible for large-scale or long-term applications [5,6]. Consequently, there is an urgent need for sustainable, cost-effective, and ecologically friendly solutions to address this growing problem.

Phytoremediation, the use of plants to remove, detoxify, or sequester pollutants from the environment, emerges as a promising alternative. Aquatic macrophytes have shown potential in absorbing and detoxifying various contaminants, including heavy metals and organic pollutants. This natural approach not only improves water quality but also enhances the ecological balance of aquatic systems [7]. Among the aquatic plants studied for their phytoremediation capacities, certain species stand out for their efficiency and adaptability. *Pistia stratiotes*, commonly known as water lettuce, and *Ceratophyllum demersum*, known as hornwort, are widely distributed and easily cultivated aquatic plants. Their rapid growth rates, high tolerance to pollutants, and capacity to bioaccumulate toxins make them ideal candidates for phytoremediation [3,4]. However, the specific mechanisms and efficiency of these plants in bioaccumulating microcystins remain underexplored. Recent research has begun to shed light on this issue, particularly by highlighting the responses of *Ceratophyllum demersum* to microcystin exposure, which appear to be similar to those observed in *Vallisneria natans*. In both aquatic macrophytes, a notable antioxidant response was triggered, marked by the activation of key enzymes such as glutathione S-transferase (GST), superoxide dismutase (SOD), and peroxidase (POD). Concurrently, elevated levels of malondialdehyde (MDA), a well-known biomarker of oxidative stress, were detected. These findings suggest that aquatic macrophytes deploy comparable defense mechanisms to mitigate the toxic effects of microcystins, although the magnitude of these responses may vary depending on species-specific traits and environmental conditions [8,9].

This study aims to fulfill this knowledge gap by providing comprehensive insights into the bioaccumulation patterns and biochemical responses of *Pistia stratiotes* and *Ceratophyllum demersum* over a 15-day period. Key physicochemical parameters, including temperature, pH, dissolved oxygen, and total dissolved solids (TDS), have been monitored to assess their influence on plant efficiency and water quality. Additionally, the activation of antioxidant defense systems in response to oxidative stress induced by microcystins was assessed, focusing on enzymes such as superoxide dismutase (SOD) and glutathione-S-transferase (GST).

## 2. Results

### 2.1. Effect of Environmental Parameters on Pistia stratiotes and Ceratophyllum demersum

Our results showed that *Pistia stratiotes* was influenced by various environmental parameters such as temperature, pH, dissolved oxygen, and total dissolved solids (TDS). Temperatures fluctuated between 24 and 28 °C, influencing the metabolic rates and stress responses of the plant. The neutral pH range of 7.13 to 7.55 facilitated better regulation of metabolic processes. Stable dissolved oxygen levels (8–12 mg/L) supported respiration and photosynthesis, while variations in TDS demonstrated efficient nutrient uptake by the roots of *Pistia stratiotes*. Environmental parameters also played a crucial role in the bioaccumulation of microcystins by *Ceratophyllum demersum*. Stable temperatures between 24 and 25 °C and a neutral pH of 7.13 to 7.55 provided optimal conditions for the plant’s metabolic processes. Dissolved oxygen levels slightly decreased from 10.32 mg/L to 8.39 mg/L, remaining within favorable ranges for respiration and photosynthesis. These stable conditions contributed to maintaining consistent bioaccumulation rates throughout the study.

### 2.2. Effects on Chlorophyll a, b, and Carotenoids

*Pistia stratiotes* exhibited significant changes in chlorophyll a, b, and carotenoid concentrations in response to microcystin exposure (Figure 1). Chlorophyll a initially decreased before showing a slight increase towards the end of exposure, indicating an adaptation to toxic stress. Additionally, carotenoid levels increased, suggesting the activation of antioxidant defense mechanisms to protect photosynthetic structures from oxidative damage. Exposure to microcystins resulted in variations in chlorophyll a and b concentrations (Figure 2), as well as carotenoids in *Ceratophyllum demersum*. A decrease in chlorophyll a was observed, with no notable recovery, indicating persistent photosynthetic stress. In contrast, carotenoid levels increased from the early days of exposure, supporting the hypothesis of an activated antioxidant response to counteract oxidative damage induced by toxins.

### 2.3. Stress Effect and Dose–Response Relationship

Microcystin exposure induced oxidative stress responses (Figure 3) in *Pistia stratiotes*, as evidenced by changes in glutathione (GSH), glutathione-s-transferase (GST), superoxide dismutase (SOD), and malondialdehyde (MDA) levels. A dose–response relationship was observed, with increasing doses of toxins leading to higher levels of GSH and MDA. GSH (*p* = 0.000) showed regeneration in individuals exposed to low doses, while MDA levels indicated effective management of oxidative stress (*p* = 0.000). Defense enzymes (GST and SOD) were strongly induced to metabolize GSH and combat MDA (*p* < 0.05) to facilitate the elimination of free radicals (ROS).

The response to oxidative stress of *Ceratophyllum demersum* was marked by significant variations in GSH, GST, SOD and MDA levels (Figure 4). A dose–response relationship was observed, with increasing doses of microcystins leading to higher GSH levels, followed by a decrease towards the end of exposure (*p* = 0.000). A dose-effect relationship was notable in *Ceratophyllum*; during the 15 days of follow-up, the GST level continued to increase with increasing dose and study period (*p* = 0.000). In contrast to *Pistia*, the superoxide dismutase level was very low compared to the control, indicating that in *Ceratophyllum*, the cellular protection system is still controlled by GSH (*p* < 0.05). MDA levels showed an increase at day 10 (*p* = 0.000), suggesting oxidative damage that was subsequently repaired during detoxification.

### 2.4. Bioaccumulation Results

The bioaccumulation index (BA) of *Pistia stratiotes* revealed high rates at the beginning of exposure (Figure 5A), with 29.21 µg/L/day for dose 1 and 36.40 µg/L/day for dose 3 on day 3. However, these rates significantly declined over time, indicating rapid saturation or metabolic limitation. This behavior suggests that *Pistia stratiotes* is effective for short-term interventions aimed at quickly reducing microcystin levels in contaminated water bodies but requires regular management to maintain its efficacy.

*Ceratophyllum demersum* maintained stable bioaccumulation rates throughout the study period (Figure 5B). Bioaccumulation indices were 43.74 µg/L/day on day 3 for dose 1 and 30.43 µg/L/day for dose 3, indicating a consistent capacity for bioaccumulation. This stability suggests that *Ceratophyllum demersum* is particularly well-suited for long-term phytoremediation projects, offering a sustainable solution for reducing microcystins in contaminated aquatic environments.

### 2.5. Comparison of the Principal Component Analysis (PCA)

The Principal Component Analysis (PCA) was conducted for both plants to reduce the dimensionality of the data while retaining significant variance explained by the original variables. For the first plant, “*Pistia*” (Figure 6A), the first two principal components explained 25.09% (PC1) and 20.71% (PC2) of the total variance, respectively. For *Ceratophyllum* (Figure 6B), the first two principal components explained 21.79% (PC1) and 19.78% (PC2) of the total variance. In both cases, the vectors of Carotenoid and Chl a showed a strong positive contribution to PC1. However, while MDA and GST were notable influencers for the first plant, SOD and GSH significantly influenced both components for *Ceratophyllum*. The samples for both plants clustered distinctly according to the days, indicating similar chemical profiles for each day. The differences in sample distribution in the principal component space were primarily explained by variations in these key variables. This comparative analysis reveals underlying relationships and similarity patterns among the samples, highlighting both shared and unique characteristics of the two plants.

## 3. Discussion

Monitoring physicochemical parameters during the study provided valuable data to assess water quality, predict environmental changes, and ensure water resource safety. Aquatic plants directly influence physical parameters such as temperature, turbidity, oxygen concentration, pH, and conductivity, which are crucial for maintaining the quality and stability of aquatic ecosystems.

The water temperatures remained stable, ranging between 24 and 25 °C for both plants across all three concentrations studied. This stability is essential for the metabolic processes of plants and aligns with findings by [10], who highlighted that submerged plants create favorable microclimates, lowering surface temperatures and benefiting temperature-sensitive aquatic species.

The close to neutral pH range (7.13 to 7.55) observed in our study is attributed to the regulatory effect of the aquatic plants, which modulate CO_2_ concentration during photosynthesis. This aligns with [11], who emphasized the role of aquatic plants in maintaining the chemical balance of aquatic ecosystems.

The observed reduction in turbidity for both plants and across all concentrations can be attributed to the plants’ ability to trap suspended particles, as suggested [12]. This natural filtration mechanism enhances water clarity and quality.

Dissolved oxygen concentration is a critical parameter for aquatic life, influencing organic matter degradation and photosynthesis processes [13]. The stable oxygen levels observed in our study indicate good aeration, corroborated by [14], who noted that macrophytes contribute dissolved oxygen through photosynthesis.

Conductivity serves as an indicator of water purity, with lower values indicating higher purity [15]. Aquatic plants’ absorption of nutrients and minerals can reduce conductivity, a sign of improved water quality [16]. In our experiments, the TDS values fluctuated from 0.202 mg/L on day 0 to 0.522 mg/L on day 15, with a notable reduction to 0.189 mg/L in the *Ceratophyllum demersum* samples, suggesting decreased water mineralization due to the presence of these plants.

Aquatic macrophytes are recognized for their bioindication capabilities, with their sensitivity to microcystins allowing for water quality monitoring and detection of toxic cyanobacterial blooms [17]. Numerous studies have confirmed the accumulation of microcystins in both terrestrial and aquatic plants, with bioaccumulation dependent on exposure duration and toxin concentrations [3,18,19].

Regarding carotenoids, an increase in their concentration can constitute an enhanced antioxidant defense mechanism, activated in response to stress conditions, to protect cellular membranes and photosynthetic structures. This phenomenon was observed in *Ceratophyllum demersum*, where an increase in carotenoids was noted during the first three days of exposure, particularly in the group treated with *Pistia stratiotes*. This increase was triggered from the beginning of the exposure, aligning with previous studies, such as those by [20], which reported increased carotenoid production after 12 h of exposure, reaching their peak after 48 h.

Furthermore, a study by [8] also indicated a significant increase in carotenoids in macrophytes *Ceratophyllum demersum* and *Egeria densa* exposed to microcystins (MC) after only 4 and 8 h, respectively. *Hydrilla verticillata* showed a similar response, but carotenoid production was delayed, occurring after 8 h to 3 days. This capacity of *Ceratophyllum demersum* to produce higher levels of carotenoids compared to other species is consistent with our findings, suggesting that this plant is particularly effective at activating its antioxidant defense mechanisms in response to microcystins.

Overall, these results show that all macrophytes exposed to microcystins appear to rely on their ability to increase antioxidant carotenoid concentration to cope with the oxidative stress induced by the toxins, confirming the importance of this response in protecting against cellular damage.

The antioxidant defense system, composed of various enzymes such as superoxide dismutase (SOD), catalase (CAT), peroxidase (POD), and GST, plays a crucial role in protecting the organism against MC toxicity [21]. The stress levels observed in our results could also be linked to the plants’ efforts to biotransform microcystins (MC), as documented in several previous studies. The biotransformation of MC is facilitated by the conjugation of glutathione (GSH) to MC conjugates via soluble GST [22,23].

Our results reveal an increase in GST levels in *Pistia stratiotes* and *Ceratophyllum demersum*, which play an important role in detoxification after exposure to microcystins (MC), with a maximum concentration detected in individuals exposed to the highest dose (5 µg/L). These observations are consistent with the work of [22], who reported an increase in GST in *Ceratophyllum demersum* after exposure to MC. Additionally, research by [24] showed a similar response in *Vallisneria natans*, where a significant increase in GST was observed following MC exposure. These results suggest that MC exposure induces a comparable adaptive GST response in various aquatic species, likely as a defense mechanism against microcystin toxicity.

Furthermore, earlier research by [25] also highlighted a significant increase in GST activity, similar to our findings. This increase in enzymatic activity indicates intensified metabolism of microcystin-LR (MC-LR) in the aquatic plants studied, notably *Ceratophyllum demersum*, *Vallisneria natans*, and *Myriophyllum spicatum*. This process confirms their capacity to partially detoxify or neutralize this toxin, enhancing their resistance and adaptation to MC-LR-contaminated environments. This potential use of these plants in phytoremediation strategies to purify polluted aquatic environments is significant.

Moreover, the study [8] demonstrated that the three macrophytes, *Ceratophyllum demersum*, *Egeria densa*, and *Hydrilla verticillata*, exhibit the capacity to rapidly absorb microcystins (MC) present in their environment. This immediate absorption appears to be an effective phytoremediation strategy, limiting MC toxicity in contaminated waters. The activation of the antioxidant defense system was highest in *Ceratophyllum* compared to other macrophytes studied. The increased GST activity in both studied macrophytes could also be interpreted as a potential indicator of microcystin absorption and subsequent biotransformation, as described in several studies on *Ceratophyllum* [3,8].

Glutathione (GSH) plays a crucial role in maintaining cell health and overall well-being due to its powerful antioxidant and detoxifying properties. It protects against damage induced by free radicals and toxins, including MC [23,26].

In our study, *Ceratophyllum demersum* exhibited variations in GSH levels, with an increase during the initial days of exposure, followed by a decrease at the end of exposure in all three groups. In contrast, *Pistia stratiotes* showed a dose-dependent relationship, with GSH levels increasing with higher toxin doses, followed by a reversal in trend. Regeneration of GSH was noted in individuals exposed to low doses, while a decrease was observed in the second group, and stabilization occurred in the third group by the end of the exposure. Our results can be explained by a decrease in GSH content, likely due to increased consumption during the formation of MC-GSH conjugates. Additionally, this conjugation stimulated new GSH synthesis, leading to an increase in GSH content, as documented in several studies [22,25,27]. Thus, the initial increase in GSH content during bioaccumulation and early exposure is likely explained by the formation of GSH conjugates via the GST system, allowing detoxification of MC [25].

An increase in malondialdehyde (MDA) levels was observed following exposure to MC extract. However, a low concentration was noted during the last days of exposure, indicating a notable detoxifying capacity in *Pistia stratiotes*, while *Ceratophyllum demersum* also demonstrated remarkable detoxifying ability at the lowest concentration (*p* < 0.01) throughout the study period. In contrast, the second and third concentration groups exhibited significant induction (*p* < 0.01) of MDA levels in plant tissues at 10 and 15 days of treatment.

The increase in MDA levels during the accumulation period indicates that MC induced oxidative damage and lipid peroxidation, suggesting that cellular membranes were damaged due to lipid peroxidation. This conclusion is supported by several studies [8,20].

Our results indicate that *Pistia stratiotes* demonstrated higher resistance to MC, with no notable negative effects at a concentration of 5 µg/L of MC. A decrease in MDA levels in *Pistia stratiotes* can be interpreted as a sign of effective management of oxidative stress by the plant, achieved through successful activation of antioxidant defense mechanisms.

During the detoxification phase, an increase in MDA content was observed in *Ceratophyllum demersum* on day 10, suggesting that oxidative damage induced by MC might be progressively repaired during detoxification. Our observations also show that glutathione (GSH) plays a crucial role in MC detoxification in *C. demersum*. These results are consistent with those reported by [25] on the same species. Additionally, an earlier study by [22] demonstrated intensified oxidative stress in *C. demersum* during the biotransformation of microcystin-LR (MC-LR).

The study revealed that superoxide dismutase (SOD), a crucial enzyme for managing oxidative stress in plants, increased in activity in response to the oxidative stress induced by the presence of toxins (MC). Higher toxin concentrations led to an increase in SOD activity. In *Pistia stratiotes*, significant fluctuations and inductions (*p* < 0.05) of SOD activity were observed after 15 days of exposure to toxic extracts, with the maximum concentration measured at 1.597 U SOD/mg of protein in plants treated with µg/L of MCs.

In *Ceratophyllum demersum*, SOD levels were very low compared to the control, with the highest concentration at the end of the study being 2.67 U SOD/mg of protein detected in *Ceratophyllum* exposed to 1 µg/L of MCs. Based on these results, superoxide dismutases (SOD) may play a role in the response to oxidative stress caused by microcystins, aligning with the work of [28], who demonstrated this interaction for microcystin-LR (MC-LR). This suggests that, in response to oxidative damage induced by microcystins, cells mobilize enzymatic defense mechanisms, such as SOD, to neutralize reactive oxygen species (ROS) and thus limit cellular damage [29].

The bioaccumulation of microcystins in macrophytes results from a highly complex process, influenced by numerous factors, both biotic—related to the specific characteristics of plant species—and abiotic—related to environmental conditions. The anatomy and physiology of the plants also play a major role in this process. Species with large leaf surfaces, fine submerged tissues, permeable structures, or a well-developed and complex root system generally exhibit a greater capacity to bioaccumulate microcystins [30].

Within this dynamic, certain floating plants can accumulate toxins both in their submerged tissues and in their emergent parts, reflecting their potential for multisource bioaccumulation [31].

In parallel, the sensitivity of plant species to microcystin plays a fundamental role in their ability to bioaccumulate this toxin. Some macrophyte species exhibit high tolerance to microcystin, enabling them to absorb and accumulate the toxin in their tissues without notable effects on their growth, often due to enzymatic detoxification mechanisms Conversely, more sensitive species show signs of physiological stress, including reduced photosynthesis and a marked slowdown in growth under high toxin concentrations [21,32].

The bioaccumulation of microcystins by aquatic macrophytes such as *Pistia stratiotes* and *Ceratophyllum demersum* is essential for understanding their potential applications in phytoremediation. This study provides comprehensive insights into the bioaccumulation patterns and biochemical responses of these plants over a 15-day period under varying environmental conditions.

*Pistia stratiotes* exhibited high initial bioaccumulation rates, particularly in the first few days of exposure, with indices of 29.21 µg/L/day for dose 1 and 36.40 µg/L/day for dose 3 on day 3. These rates, however, declined significantly over time, indicating a rapid saturation or metabolic limitation, which aligns with findings from [33,34]. This rapid initial uptake suggests *Pistia stratiotes* is effective for short-term interventions in reducing microcystin levels in water bodies.

Conversely, *Ceratophyllum demersum* maintained more stable bioaccumulation rates, with indices of 43.74 µg/L/day on day 3 and 30.43 µg/L/day for doses 1 and 3, respectively. This stability suggests a consistent capacity for long-term remediation, as supported by studies from [35].

The results underscore the importance of selecting appropriate aquatic plants based on specific remediation needs. *Pistia stratiotes* is suitable for rapid initial detoxification, but its effectiveness diminishes over time, necessitating periodic harvesting and replacement. On the other hand, *Ceratophyllum demersum* offers sustained bioaccumulation capabilities, making it ideal for long-term phytoremediation projects.

In practical applications, integrating both plants could optimize microcystin removal from contaminated water bodies. *Pistia stratiotes* can be used for immediate intervention, while *Ceratophyllum demersum* provides a stable, long-term solution. This combined approach could enhance the overall efficiency and sustainability of bioremediation programs, contributing to better water management and public health protection.

## 4. Conclusions

This study highlights the efficiency of *Pistia stratiotes* and *Ceratophyllum demersum* in the bioaccumulation of microcystins, presenting a promising phytoremediation method for managing cyanotoxins in aquatic ecosystems. The results demonstrate not only the plants’ ability to absorb and detoxify microcystins but also the significance of environmental parameters in optimizing this process. Moreover, these studies provide a better understanding of the growth, accumulation and degradation processes of cyanobacterial strains and pave the way for the development of analytical sensing systems that monitor cyanobacterial toxins in the environment. Applying these techniques in reservoirs and other water bodies could provide an eco-friendly and sustainable solution for reducing cyanotoxin levels, thereby improving water quality and protecting aquatic ecosystems. By integrating these natural treatment methods, reservoir managers can effectively control cyanobacterial blooms, reduce the costs associated with chemical treatments, and minimize negative environmental impacts. These findings encourage the broader adoption of phytoremediation in water management policies, offering a pathway towards more environmentally friendly and sustainable practices for treating waters contaminated with cyanotoxins. Ultimately, selecting the most appropriate phytoremediation strategy should be guided by specific bioremediation goals, the nature of the contaminants, and the ecological context of the aquatic system.

## 5. Material and Methods

### 5.1. Cyanobacteria Bloom Samples

Cyanobacterial bloom samples, dominated by the species *Microcystis* sp., were collected from Lake des Oiseaux, North-East Algeria [36]. Microcystin variants were characterized using Ultra-High-Performance Liquid Chromatography coupled with tandem mass spectrometry (UHPLC-MS/MS). The analysis was performed on a Waters Acquity UPLC system linked to a triple-quadrupole mass spectrometer (TQD, Waters, Paris, France) via an electrospray ionization interface operating in positive mode. Separation was carried out on a BEH C18 column (100 × 2.1 mm, 1.7 µm particle size) maintained at 30 °C. A linear gradient elution was applied at a flow rate of 0.3 mL/min using MilliQ water and acetonitrile, both containing 0.1% formic acid. The acetonitrile concentration increased progressively: 22.5% to 40.5% at 4 min; 75% at 10 min; and 100% at 11 min, at which it was held for 1 min. The injection volume was set at 10 µL. Electrospray parameters included a capillary voltage of 3 kV, cone voltage of 50 V, source temperature of 120 °C, and desolvation temperature of 300 °C. Nitrogen was used as both the cone and desolvation gas at flow rates of 20 L/h and 800 L/h, respectively. Mass spectra were acquired across a range of 100–1200 Da. For quantification, ion currents corresponding to [M+H]^+^ (and [M + H]^+^ + [M + 2H]^2+^ for RR variants) were monitored. Tandem MS analysis was conducted via Collision-Induced Dissociation (CID) using argon at 3.4 mbar, with collision energies of 50 and 70 eV. Parent ion scanning targeted *m*/*z* 135, while daughter ion scans focused on fragmentation of the [M + H]^+^ ions. The total microcystin content per phytoplankton biomass was estimated in respect to most toxic MC-LR variant amount, and was 0.062 mg MC-LR equivalents/g dried bloom material, with 21 potential microcystin variants, among them, some variants were already extensively reported in the literature: MC-RR (43.4% of the total), MC-LR (13.1%), MC-FR (9.4%), MC-YR (5.4%) and MC-WR (7.4%) (Figure 7) [37,38,39,40,41,42].

### 5.2. Experiment Design

Our study was carried out on aquatic macrophytes, *Pistia stratiotes* and *Ceratophyllum demersum*, supplied by an aquatic farm in Algiers, Algeria. The plants were acclimatized under laboratory conditions at 25 °C average temperature, 12/12 h of photoperiod and pH 7. The robust apical sections of the young *C. demersum* and *P. stratiotes* plants, 20 cm in length, were thoroughly cleaned with distilled water to remove any attached debris present and pre-cultured for 10 days in aquaria (25 cm × 12 cm × 15 cm) to ensure their adaptation to the laboratory environment.

After the adaptation period, both plants were treated as follows:

Batch N°1: control individuals with the addition of 1 mL of physiological water.

Batch N°2: individuals with the addition of 1 mL of physiological water containing 1 µg/L equivalent MC-LR of *Microcystis* bloom.

Batch N°3: individuals with the addition of 1 mL of physiological water containing 2.5 µg/L equivalent MC-LR of *Microcystis* bloom.

Batch N°4: individuals with the addition of 1 mL of physiological water containing 5 µg/L equivalent MC-LR of *Microcystis* bloom.

All experiments were performed in triplicate and under identical conditions for each group. After 15 days of exposure to *Microcystis* bloom, the plants were cleaned with distilled water and cut into biopsies: a sample of 100 mg was placed in a phosphate buffer (pH 7.4) and was further used for dosage determination of stress biomarkers. Other samples were used to assess the dosage of biological parameter rates and detection of bioaccumulation rates; the remaining plants were frozen immediately and stored at −20 °C to be reutilized in case of necessity. The biological parameters and bioaccumulation parameters were analyzed during a 15-day experiment period with sampling performed in triplicate on the 3rd, 5th, 10th and 15th day, and were accompanied by the detection of environmental parameters of the plant breeding environment.

### 5.3. Environmental Parameters

On the 1st, 3rd, 5th, 10th and 15th days, water quality parameters were detected using a benchtop multiparameter (HORIBA, water quality monitor), including temperature, pH, dissolved oxygen, TDS, salinity, turbidity, and conductivity.

### 5.4. Measurements of Biological Parameters

The levels of chlorophyll a, chlorophyll b, and carotenoids were quantified to assess the impact of cyanotoxin exposure on plant pigmentation and photosynthetic activity. Chlorophylls a and b were extracted and analyzed following the method described by [43]. All extractions were conducted in the dark to prevent pigment degradation.

For each extraction, 0.2 g of the plant material was homogenized in 0.5 mL of 90% acetone using a mortar and pestle, while kept cool on ice. The homogenate was then transferred to a microcentrifuge tube and centrifuged for 5 min at 10,000× *g*. The resulting pellet was extracted twice with 0.5 mL of 90% acetone.The supernatants from each extraction were combined, and absorbance was measured with a Jenway 7310 spectrophotometer (Thermo Fisher Scientific) at specific wavelengths: 664 nm and 647 nm for chlorophylls, and 470 nm for carotenoids [44].

### 5.5. Oxidative Stress Biomarkers

Frozen tissue aliquots were thawed on crushed ice and homogenized with 10 mM ice-cold phosphate buffer (pH 7.4) using a manual homogenizer. The homogenate was then vortexed and centrifuged at 10,000× *g*/10 min at 4 °C. Aliquots of each supernatant were used to measure total protein content and oxidative stress biomarkers using spectrophotometric methods. Protein content was determined using the Bradford dye-binding assay with bovine serum albumin as the standard [45].

Glutathione (GSH) content was quantified using Elman’s reagent (DTNB) as described by [46]. The reaction mixture contained 500 µL of the supernatant, 1.0 mL of Tris-EDTA buffer (0.02 M, pH 9.6), and 25 µL of DTNB (0.01 M). After thorough mixing and incubation at room temperature for 5 min, fluorescence at 412 nm was measured. GSH concentration values were calculated from a pure GSH standard curve and expressed as nanomoles of glutathione per milligram of protein (nmol GSH/mg protein).

Glutathione S-transferase (GST) activity was measured using 1-chloro-2,4-dinitrobenzene (CDNB) as a substrate, following the method of [47]. The reaction mixture contained 50 µL of tissue supernatant, 1.05 mL of 100 mM TRIS buffer (pH 7.4), 50 µL of 1 mM CDNB, and 50 µL of 1 mM GSH. GST activity was determined by monitoring changes in absorbance at 340 nm for 2 min, reflecting the rate of CDNB conjugation with GSH, using an extinction coefficient of 9.6 mM^−1^ cm^−1^. One unit (U) of GST activity is defined as the amount of enzyme that conjugates 1 µmole of CDNB per minute, and GST activity is expressed as U/mg protein.

The Marklund and Marklund method is based on the ability to inhibit the autoxidation of pyrogallol in the presence of EDTA by SOD. The assay was carried out in a final volume of 1 mL. To 850 μL of Tris HCl buffer (50 mM, pH: 8.2) were added 20 μL of the sample, 100 μL of EDTA and 50 μL of pyrogallol (2.5 mM prepared in 10 mM of HCl). The change in absorbance was measured at 420 nm after every minute over a 5-min time interval. The enzyme activity was expressed in units/mg of protein [48].

Lipid peroxidation (LPO) levels were assessed by measuring malondialdehyde (MDA), a decomposition product of polyunsaturated fatty acids hydroperoxides, using the thiobarbituric acid (TBA) reaction as described by [49] with modifications by [50]. The assay mixture contained 1 mL of tissue supernatant, 1 mL of 5% trichloroacetic acid (TCA), and 1 mL of 0.67% TBA. The mixture was heated at 95 °C for 40 min, cooled, and centrifuged at 3000× *g* for 10 min. The absorbance of the supernatant was measured at 532 nm. The amount of thiobarbituric acid reactive substances (TBARS) was calculated using an extinction coefficient of 1.56 × 10^5^ M^−1^ cm^−1^. Lipid peroxidation rate was expressed as nanomoles of TBARS formed per hour per milligram of protein (nmol TBARS/mg protein).

### 5.6. Extraction, Determination, and Analysis of Microcystin Concentration

#### 5.6.1. Detection and Quantification of Microcystins in Plant Cells

The quantification of microcystin concentrations in plants was performed according to the method by [25], with modifications. Fresh plant tissue (0.1 g), taken from the entire plant, was ground and mixed with 2 mL of 70% aqueous methanol (*v*/*v*). The homogenate was subjected to an ultrasonic bath for 5 min, then centrifuged at 12,000 RPM at 4 °C for 15 min to remove cellular debris. It was then homogenized with a total of 13 mL of 70% aqueous methanol (*v*/*v*). The supernatant was collected and applied to a C18 Bakerbond SPD cartridge (3 mL (200 mg), particle size 45 µm), preconditioned with 100% methanol and distilled water. The microcystin in the cartridge was eluted with 20% methanol, and the eluate was dried with 3 mL of methanol acidified with 0.1% trifluoroacetic acid (TFA) to dissolve the toxin. The eluate was evaporated using a Rotavapor at 40 °C, and the dry residue was redissolved with 1 mL of 50% methanol in an Eppendorf tube for HPLC measurement.

#### 5.6.2. Calculation of Bioaccumulation Rate

The concentration of microcystins in the plant tissues was measured at different time intervals during the experiment. The bioaccumulation rate (BA) was calculated by assessing the increase in microcystin concentration over time within the plant tissues according to the following equation:BA = (C_t_ − C_0_)/t
where C_t_ is the concentration of microcystins in the plant tissues at time t; C_0_ is the initial concentration of microcystins in the plant tissues; and t is the time interval between the initial and final measurements.

This method allows for the determination of the efficiency and capacity of the plants to accumulate microcystins over time. The results were expressed as µg/L/day, representing the change in microcystin concentration in the surrounding water per day of exposure. This reflects the dynamic reduction in toxins from the medium due to plant uptake.

This approach provides a clear understanding of the bioaccumulation dynamics and helps in comparing the effectiveness of different plant species in removing microcystins from the water.

### 5.7. Statistical Analysis

All analyses were performed using IBM SPSS Statistics 25 software. The Shapiro–Wilk test was used to check whether variables followed a normal distribution and were displayed as mean ± SE. Group means were compared using one-way ANOVA when ANOVA was significant, and multiple comparisons of mean values were separated by Tukey’s test. The correlation coefficients for normal variables were calculated by Pearson’s correlation test. Statistical differences were determined at the *p* < 0.05 and *p* < 0.01 levels for all analyses.

## Figures and Tables

**Figure 1 toxins-17-00490-f001:**
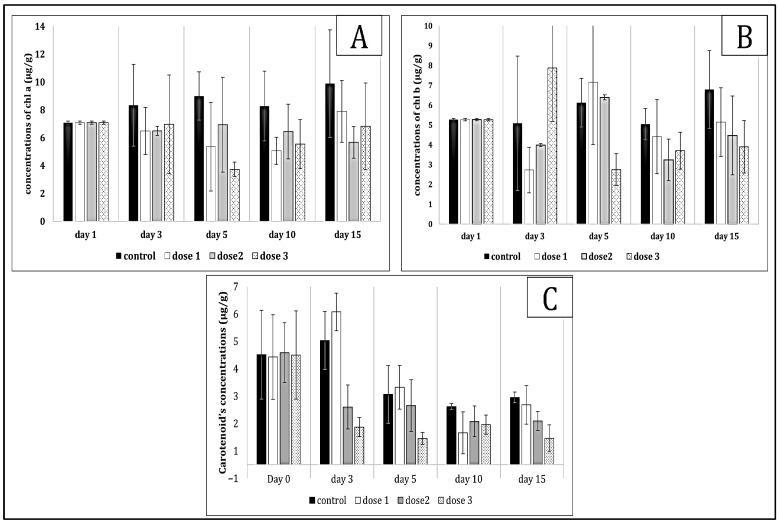
Chl a (**A**), b (**B**) and carotenoid (**C**) values in *Pistia stratiotes* tissue exposed to *Microcystis* sp. bloom. The values are expressed as mean ± S.E. The number of measurements performed in each group was three.

**Figure 2 toxins-17-00490-f002:**
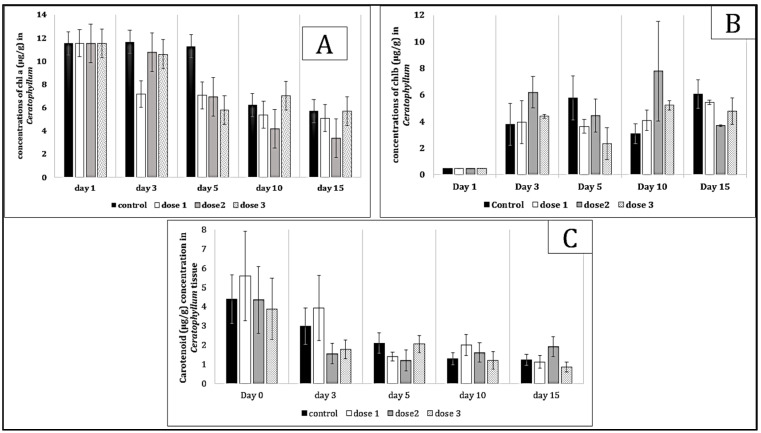
Chl a (**A**), b (**B**) and carotenoid (**C**) values in *Ceratophyllum demersum* tissue exposed to *Microcystis* sp. bloom. The values are expressed as mean ± S.E. The number of measurements performed in each group was three.

**Figure 3 toxins-17-00490-f003:**
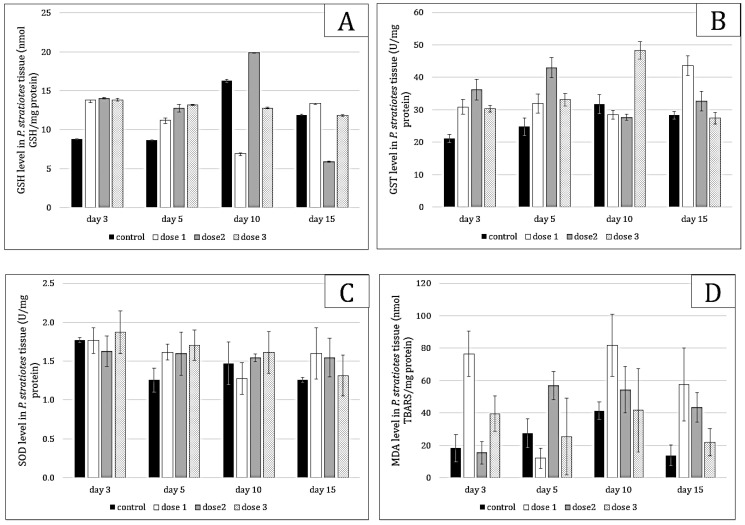
GSH (**A**), GST (**B**), SOD (**C**) and MDA (**D**) values in *Pistia stratiotes* tissue exposed to *Microcystis* sp. bloom. The values are expressed as mean ± S.E. The number of measurements performed in each group was three.

**Figure 4 toxins-17-00490-f004:**
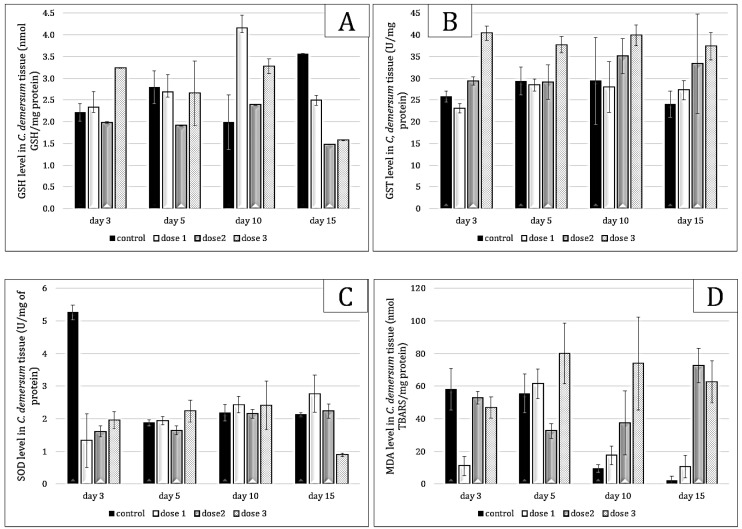
GSH (**A**), GST (**B**), SOD (**C**) and MDA (**D**) values in *Ceratophyllum demersum* tissue exposed to *Microcystis* sp. bloom. The values are expressed as mean ± S.E. The number of measurements performed in each group was three.

**Figure 5 toxins-17-00490-f005:**
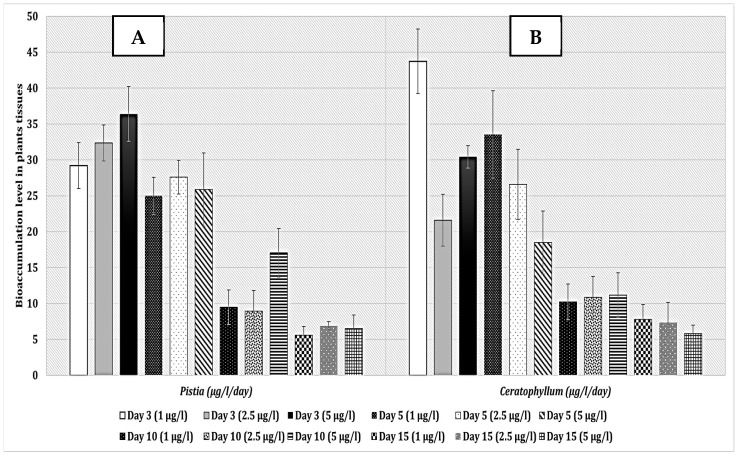
Bioaccumulation values in *Pistia stratiotes* tissue (**A**) and *Ceratophyllum demersum* (**B**) exposed to *Microcystis* sp. bloom. The values are expressed as the mean. The number of measurements performed in each group was three.

**Figure 6 toxins-17-00490-f006:**
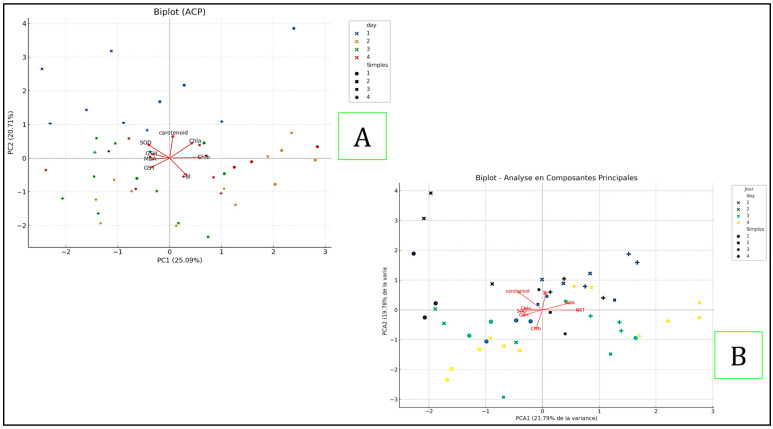
The Principal Component Analysis (PCA) for *Pistia stratiotes* (**A**) and *Ceratophyllum demersum* (**B**).

**Figure 7 toxins-17-00490-f007:**
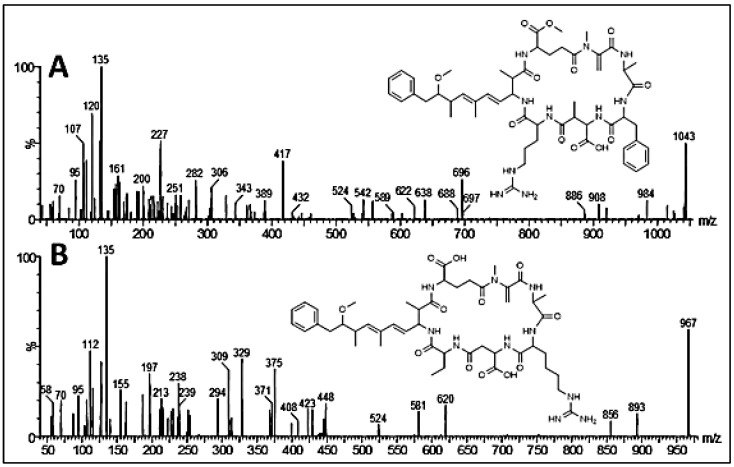
CID mass spectra of [MþH]þ ions of the two new microcystin variants: (**A**) [Glu(OCH_3_)^6^]MC-FR, [MþH]þ 1043; (**B**) [Asp3]MC-HarAba, [MþH]þ 967. Inserts show the chemical structure of the new MC variants [36].

## Data Availability

The original contributions presented in this study are included in the article. Further inquiries can be directed to the corresponding authors.

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
