# Peer review of "Bioremediation of Contaminated Water: The Potential of Aquatic Plants Ceratophyllum demersum and Pistia stratiotes Against Toxic Bloom"

_toxins, 2025, doi:10.3390/toxins17100490_

Round 1
Reviewer 1 Report
Comments and Suggestions for Authors
This study aims to investigate bioaccumulation of microcystins by two macrophytes to evaluate their potential as phytoremediatory agents for harmful cyanobacterial blooms. The main findings of the article are that both C. demersum and P. statiotes both exhibited phytoremediation potential with C. demersum being more stable in terms of bioaccumulation and P. statiotes exhibiting higher initial rates. The key conclusion is that while both macrophytes are effective at phytoremediation of microcystins, the choice of which to use is dependent on the bioremediation needs. This study is very significant in the field of phytoremediation and of wide interest to the readers of Toxins. Here are some revisions that would help the article.
Comments:
- For the UHPLC-MS/MS analysis, please provide CID conditions and please share the mzXML or mzML data for this study in a repository. Given the data collected, the MassIVE repository is a good option, but any repository would be fine. https://massive.ucsd.edu/ProteoSAFe/static/massive.jsp?redirect=auth
- Please make sure that scientific names are italicized. Examples where this is currently not the case are line 106 and Figure 2.
- Please ensure that figures are high enough resolution that they are not blurry. Also, please make sure to label x and y axes as needed.
- Please make sure that standard errors and/or standard deviations are reported in all of your figures. They are missing in figure 4.
- It would be beneficial to add one more sentence to your conclusions talking about choosing the phytoremediation method that works best for the bioremediation needs.
- Please make sure references include the DOIs.
Author Response
Reviewer #1: This study aims to investigate bioaccumulation of microcystins by two macrophytes to evaluate their potential as phytoremediatory agents for harmful cyanobacterial blooms. The main findings of the article are that both C. demersum and P. statiotes both exhibited phytoremediation potential with C. demersum being more stable in terms of bioaccumulation and P. statiotes exhibiting higher initial rates. The key conclusion is that while both macrophytes are effective at phytoremediation of microcystins, the choice of which to use is dependent on the bioremediation needs. This study is very significant in the field of phytoremediation and of wide interest to the readers of Toxins. Here are some revisions that would help the article.
Responses of authors:
Thank you so much for your comments. We appreciate it.
- For the UHPLC-MS/MS analysis, please provide CID conditions and please share the mzXML or mzML data for this study in a repository. Given the data collected, the MassIVE repository is a good option, but any repository would be fine. https://massive.ucsd.edu/ProteoSAFe/static/massive.jsp?redirect=auth
Responses of authors:
- We thank the reviewer for this valuable suggestion regarding data transparency and reproducibility.
The matrix used in this study is part of an ongoing international research collaboration between Algerian and European institutions. Due to consortium agreements and confidentiality clauses associated with this project, the detailed composition of the matrix cannot be disclosed at this stage. However, all analytical procedures, instrumentation parameters, and resulting data (including UHPLC-MS/MS profiles) have been fully described and made available to ensure transparency and reproducibility of the study. To support the reviewer’s request for additional context, we have attached the original article from the collaborative project, which provides further details on the matrix composition, sampling strategy, and preliminary results. This reference complements the current study and clarifies the broader framework in which our work is embedded.
- Please make sure that scientific names are italicized. Examples where this is currently not the case are line 106 and Figure 2.
Responses of authors:
We thank the reviewer for pointing out the formatting inconsistency regarding scientific names. We have carefully reviewed the manuscript and ensured that all scientific names are now properly italicized, including those on line 106 and in Figure 2. This correction has been applied throughout the text and figures to maintain consistency with scientific standards
- Please ensure that figures are high enough resolution that they are not blurry. Also, please make sure to label x and y axes as needed.
Responses of authors:
We thank the reviewer for highlighting the importance of figure clarity and axis labeling. All figures have been revised to ensure high-resolution quality suitable for publication. The results part is rechecked as follows:
Results
3.1. Effect of Environmental Parameters on Pistia stratiotes and Ceratophyllum demersum
Our results showed that Pistia stratiotes was influenced by various environmental parameters such as temperature, pH, dissolved oxygen, and total dissolved solids (TDS). Temperatures fluctuated between 24 and 28 °C, influencing the metabolic rates and stress responses of the plant. The neutral pH range of 7.13 to 7.55 facilitated better regulation of metabolic processes. Stable dissolved oxygen levels (8-12 mg/l) supported respiration and photosynthesis, while variations in TDS demonstrated efficient nutrient uptake by the roots of Pistia stratiotes. These environmental parameters played a key role in the plant's bioaccumulation and detoxification capacity.
Environmental parameters also played a crucial role in the bioaccumulation of microcystins by Ceratophyllum demersum. Stable temperatures between 24 and 25 °C and a neutral pH of 7.13 to 7.55 provided optimal conditions for the plant's metabolic processes. Dissolved oxygen levels slightly decreased from 10.32 mg/l to 8.39 mg/l, remaining within favorable ranges for respiration and photosynthesis. These stable conditions contributed to maintaining consistent bioaccumulation rates throughout the study, highlighting the robustness of Ceratophyllum demersum against environmental variations.
3.2. Effects on Chlorophyll a, b, and Carotenoids
Pistia stratiotes exhibited significant changes in chlorophyll a, b, and carotenoid concentrations in response to microcystin exposure (Figure 2). Chlorophyll a initially decreased before showing a slight increase towards the end of exposure, indicating an adaptation to toxic stress. Additionally, carotenoid levels increased, suggesting the activation of antioxidant defense mechanisms to protect photosynthetic structures from oxidative damage. These results underscore the ability of Pistia stratiotes to adapt and maintain its photosynthetic activity despite stressful conditions.
Exposure to microcystins resulted in variations in chlorophyll a and b concentrations (Figure 3), as well as carotenoids in Ceratophyllum demersum. A decrease in chlorophyll a was observed, with no notable recovery, indicating persistent photosynthetic stress. In contrast, carotenoid levels increased from the early days of exposure, supporting the hypothesis of an activated antioxidant response to counteract oxidative damage induced by toxins. These biochemical adjustments demonstrate the ability of Ceratophyllum demersum to adapt its defense mechanisms to environmental stress.

Reviewer 2 Report
Comments and Suggestions for Authors
Dear Authors,
I have read your manuscript entitled “Bioremediation of Contaminated Waters: The Potential of Aquatic Plants Ceratophyllum demersum and Pistia stratiotes Against Toxic Bloom”. The topic is relevant and timely, and your study provides valuable experimental evidence of the complementary roles of Pistia stratiotes and Ceratophyllum demersum in microcystin removal. The experimental design is sound, the results are clearly presented, and the discussion is well supported by the literature. However, before acceptance, I recommend minor revisions to improve clarity and consistency: 1. Figures and legends: please revise the numbering order (currently inconsistent, e.g., Fig. 2 → Fig. 5 → Fig. 3) and ensure that all axes include units. Clarify whether Fig. 1 reproduces Bouhaddada et al. (2016) or is original data. 2. References: remove the duplicate entry of Singh et al. (2023); correct the reference Weckberker & Cory (1988) (Cancer Letters); and harmonize pigment references (Lichtenthaler vs. Wellburn). 3. Methodology: clarify whether exposure doses (section 2.2) are expressed as µg/L of total bloom material or as MC-LR equivalents; and specify the units in the bioaccumulation rate formula (section 2.6.2). 4. Language: minor corrections are needed (e.g., “playsa” → “play a”, “global changing” → “global change”). Please also reduce redundancies between Results and Discussion. Once these adjustments are made, the manuscript will be suitable for publication. Sincerely,Author Response
Reviewer #2: I have read your manuscript entitled “Bioremediation of Contaminated Waters: The Potential of Aquatic Plants Ceratophyllum demersum and Pistia stratiotes Against Toxic Bloom”. The topic is relevant and timely, and your study provides valuable experimental evidence of the complementary roles of Pistia stratiotes and Ceratophyllum demersum in microcystin removal. The experimental design is sound, the results are clearly presented, and the discussion is well supported by the literature. However, before acceptance, I recommend minor revisions to improve clarity and consistency.
Responses of authors:
We sincerely thank Reviewer 2 for the encouraging and constructive feedback. We are pleased that the relevance, experimental design, and clarity of our study were appreciated. In response to the reviewer’s recommendation for minor revisions to improve clarity and consistency, we have carefully reviewed the manuscript and implemented the following changes.
- Figures and legends: please revise the numbering order (currently inconsistent, e.g., Fig. 2 → Fig. 5 → Fig. 3) and ensure that all axes include units. Clarify whether Fig. 1 reproduces Bouhaddada et al. (2016) or is original data.
Responses of authors:
We thank the reviewer for this careful observation regarding figure consistency and clarity. We have revised the numbering of all figures to follow a logical and sequential order throughout the manuscript (i.e., Fig. 1 → Fig. 2 → Fig. 3, etc.).
Additionally, we have ensured that all axes in the figures are now properly labeled with units and descriptive titles, in accordance with scientific standards. Regarding Figure 1, we confirm that it reproduces data originally published by Bouhaddada et al. (2016), and this has now been clearly stated in the figure legend, the original article is attached as an appendix
- References: remove the duplicate entry of Singh et al. (2023); correct the reference Weckberker & Cory (1988) (Cancer Letters); and harmonize pigment references (Lichtenthaler vs. Wellburn).
Responses of authors:
We thank the reviewer for the careful review of our reference list and for pointing out these inconsistencies. We have made the following corrections:
- The duplicate entry of Singh et al. (2023) has been removed.
-The reference to Weckberker & Cory (1988) has been corrected to reflect the accurate journal title and citation details: Cancer Letters, volume, pages, and DOI.
Weckberker, G., G., & Cory, J.G. (1988). Ribonucleotide reductase activity and growth of glutathione-depleted mouse leukemia L1210 cells in vitro. Cancer Letters, 40(3), 257-264. DOI: https://doi.org/10.1016/0304-3835 (88)90084-5.
-Pigment references have been harmonized. We now consistently cite Lichtenthaler (1987) for chlorophyll and carotenoid quantification, and have removed redundant or conflicting citations to Wellburn where appropriate
- Methodology: clarify whether exposure doses (section 2.2) are expressed as µg/L of total bloom material or as MC-LR equivalents; and specify the units in the bioaccumulation rate formula (section 2.6.2).
Responses of authors:
- We thank the reviewer for this insightful comment regarding dose specification and unit clarity. In Section 2.2, we have clarified that the exposure doses are expressed as µg/L of MC-LR equivalents, based on quantification of microcystin-LR using UHPLC-MS/MS. This ensures consistency with toxicological standards and facilitates comparison with other studies.
- In Section 2.6.2, we confirm that the bioaccumulation rate was expressed as µg/L/day, representing the change in microcystin concentration in the surrounding water per day of exposure. This reflects the dynamic reduction of toxins from the medium due to plant uptake.
2.6.2. Calculation of Bioaccumulation Rate
The concentration of microcystins in the plant tissues was measured at different time intervals during the experiment. The bioaccumulation rate (BA) was calculated by assessing the increase in microcystin concentration over time within the plant tissues according to the following equation:
BA = (Ct-C0)/t
where: Ct - is the concentration of microcystins in the plant tissues at time t, C0 - is the initial concentration of microcystins in the plant tissues, t - is the time interval between the initial and final measurements.
This method allows for the determination of the efficiency and capacity of the plants to accumulate microcystins over time. The results were expressed as µg/L/day, representing the change in microcystin concentration in the surrounding water per day of exposure. This reflects the dynamic reduction of toxins from the medium due to plant uptake.
This approach provides a clear understanding of the bioaccumulation dynamics and helps in comparing the effectiveness of different plant species in removing microcystins from the water.
- Language: minor corrections are needed (e.g., “playsa” → “play a”, “global changing” → “global change”). Please also reduce redundancies between Results and Discussion..
Responses of authors:
We thank the reviewer for highlighting these language issues and structural redundancies. We have carefully revised the manuscript to correct typographical errors and improve phrasing.
Additionally, we have reviewed the Results and Discussion sections to reduce repetition. Key findings are now presented concisely in the Results, while broader interpretation and contextualization are reserved for the Discussion. These adjustments enhance the clarity and flow of the manuscript.

Round 2
Reviewer 1 Report
Comments and Suggestions for Authors
Thank you to the authors for great revisions to the manuscript. The only recommendation that I have is to add a little bit more detail from the original article about the UHPLC-MS/MS method by explicitly stating the collision energy used for CID analysis (i.e. XX NCE). With that one change, the article will be publishable in my opinion.
Author Response
Reviewer #1: Thank you to the authors for great revisions to the manuscript. The only recommendation that I have is to add a little bit more detail from the original article about the UHPLC-MS/MS method by explicitly stating the collision energy used for CID analysis (i.e. XX NCE). With that one change, the article will be publishable in my opinion.
Response:
We sincerely thank the reviewer for the positive feedback and for recognizing the improvements made to the manuscript. We appreciate the final recommendation regarding the UHPLC-MS/MS method and have addressed it accordingly.
As suggested, we have now explicitly stated the collision energies used for CID analysis (50 and 70 eV) in Section 2.1 of the Methods, lines 89 -104. This addition ensures full transparency of the UHPLC-MS/MS procedure.
Cyanobacteria Bloom Samples
Cyanobacterial bloom samples, dominated by the species Microcystis sp., were collected from Lake des Oiseaux, north-East Algeria (Bouhaddada et al., 2016), Microcystin variants were characterized using Ultra-High-Performance Liquid Chromatography coupled with tandem mass spectrometry (UHPLC-MS/MS). The analysis was performed on a Waters Acquity UPLC system linked to a triple-quadrupole mass spectrometer (TQD, Waters, France) via an electrospray ionization interface operating in positive mode. Separation was carried out on a BEH C18 column (100 × 2.1 mm, 1.7 µm particle size) maintained at 30 °C. A linear gradient elution was applied at a flow rate of 0.3 mL/min using MilliQ water and acetonitrile, both containing 0.1% formic acid. The acetonitrile concentration increased progressively: 22.5% to 40.5% at 4 minutes, 75% at 10 minutes, and reached 100% at 11 minutes, held for 1 minute. The injection volume was set at 10 µL. Electrospray parameters included a capillary voltage of 3 kV, cone voltage of 50 V, source temperature of 120 °C, and desolvation temperature of 300 °C. Nitrogen was used as both the cone and desolvation gas at flow rates of 20 L/h and 800 L/h, respectively. Mass spectra were acquired across a range of 100–1200 Da. For quantification, ion currents corresponding to [M+H]+ (and [M+H]+ + [M+2H]2+ for RR variants) were monitored. Tandem MS analysis was conducted via Collision-Induced Dissociation (CID) using argon at 3.4 mbar, with collision energies of 50 and 70 eV. Parent ion scanning targeted m/z 135, while daughter ion scans focused on fragmentation of the [M+H]+ ions. The total microcystin content per phytoplankton biomass was estimated in respect to most toxic MC-LR variant amount, and was 0.062 mg MC-LR equivalents/g dried bloom material, with 21 potential microcystin variants, among them, some variants were already extensively reported in the literature: MC-RR (43.4% of the total), MC-LR (13.1%), MC-FR (9.4%), MC-YR (5.4%) and MC-WR (7.4%) (Figure 1 ; Namikoshi et al.,1992; Yuan et al., 1999; Zweigenbaum et al., 2000; Okello et al., 2010; Miles et al., 2013, 2014; Puddick et al., 2014).
Reference; https://doi.org/10.1016/j.envpol.2016.06.055